# Pre-exposure prophylaxis (PrEP) awareness, use, and discontinuation among Lake Victoria fisherfolk in Uganda: A cross-sectional population-based study

Kauthrah Ntabadde[1], Joseph Kagaayi[2,3], Victor Ssempijja[2,4], Xinyi Feng[5], Robert Kairania[2], Joseph Lubwama[2], Robert Ssekubugu[2], Ping Teresa Yeh[6], Joseph Ssekasanvu[1], Aaron A. R. Tobian[5], Caitlin E. Kennedy[6], Lisa A. Mills[7], Stella Alamo[7], Philip Kreniske[8], John Santelli[9], Lisa J. Nelson[7], Steven J. Reynolds[2,10,11], Larry William Chang[2,10], Gertrude Nakigozi[2], Mary Kate Grabowski[1,2,5]*

1 Bloomberg School of Public Health, Johns Hopkins University, Baltimore, Maryland, United States of America, 2 Rakai Health Sciences Program, Kalisizo, Uganda, 3 Makerere University School of Public Health, Kampala, Uganda, 4 Clinical Monitoring Research Program Directorate, Frederick National Laboratory for Cancer Research, Frederick, Maryland, United States of America, 5 Department of Pathology, School of Medicine, Johns Hopkins University, Baltimore, Maryland, United States of America, 6 Department of International Health, Bloomberg School of Public Health, Johns Hopkins University, Baltimore, Maryland, United States of America, 7 Division of Global HIV & TB, United States Centers for Disease Control and Prevention - Uganda, Kampala, Uganda, 8 Department of Community Health and Social Sciences, Graduate School of Public Health and Health Policy, City University of New York, New York, New York, United States of America, 9 Department of Population and Family Health and Pediatrics, Mailman School of Public Health, Columbia University, New York, New York, United States of America, 10 Division of Infectious Diseases, School of Medicine, Johns Hopkins University, Baltimore, Maryland, United States of America, 11 Division of Intramural Research, National Institute of Allergy and Infectious Diseases, National Institutes of Health, Bethesda, Maryland, United States of America

* mgrabow2@jhu.edu

## Abstract

There is limited population-level data on the pre-exposure prophylaxis (PrEP) care continuum in eastern Africa. Here, we assessed the PrEP care continuum following PrEP rollout in a Ugandan community with ~40% HIV seroprevalence. We used cross-sectional population-based data collected between September 3 and December 19, 2018 from a Lake Victoria fishing community in southern Uganda to measure levels of self-reported PrEP awareness, ever-use, and discontinuation following 2017 PrEP rollout via a U.S. President's Emergency Plan for AIDS Relief (PEPFAR)-supported phased implementation program. Our analysis included HIV-seronegative persons reporting having ever received an HIV test result. We examined associations between demographic, behavioral, and health utilization factors with each outcome using age-adjusted modified Poisson regression. There were 1,401 HIV-seronegative participants, of whom 1,363 (97.3%) reported ever receiving an HIV test result. Median age was 29 years (IQR: 23–36), and 42.3% (n = 577) were women. Most (85.5%; n = 1,166/1363) participants reported PrEP awareness, but few (14.5%; n = 197/1363) reported ever using PrEP. Among 47.7% (375/786) of men and 29.3%

**Data availability statement:** For ethical reasons related to sharing a data set containing potentially identifying or sensitive information, we do not share the full dataset underlying these analyses. Data requests may be sent to the Rakai Health Sciences Program data management office (datarequests@rhsp.org), where data are archived across all the various projects run by the RHSP (original paper forms from the Rakai Community Cohort Study surveys, as well as the electronic datasets for each survey round).

**Funding:** This study was supported by the National Institute of Allergy and Infectious Diseases (R01AI143333 to LC, Division of Intramural Research no grant number to SJR), the National Institute of Child Health and Human Development (R01HD091003 to JS), National Institute of Mental Health (R01MH115799 to LC), the Johns Hopkins University Center for AIDS Research (P30AI094189 to not applicable), and the U.S. President's Emergency Plan for AIDS Relief (PEPFAR) through the Centers for Disease Control and Prevention (NU2GGH000817 to JK). KN received training and support from National Institutes of Health Fogarty International Center (D43TW010557 to LC). The findings and conclusions in this report are those of the author(s) and do not necessarily represent the official position of the Centers for Disease Control and Prevention. The funders had no role in study design, data collection and analysis, decision to publish, or preparation of the manuscript.

**Competing interests:** The authors have declared that no competing interests exist.

(169/577) of women PrEP-eligible at time of survey, 18.9% (n = 71/375) and 27.8% (n = 47/169) reported ever using PrEP, respectively. Over half (52.3%, n = 103/197) of those who had ever used PrEP, self-reported current use. In this Lake Victoria fishing community, there were low levels of PrEP use despite high levels of PrEP awareness and eligibility, particularly among men. Efforts that enhance awareness of HIV risk and increase PrEP accessibility may help increase PrEP use among HIV-seronegative persons in African settings with high HIV burden.

## Introduction

Despite major advances in the treatment and prevention of HIV, the virus remains a major public health threat, particularly in sub-Saharan Africa [1]. Within Africa, the HIV epidemic is concentrated in the eastern and southern regions, which accounted for more than half of all new infections worldwide in 2020 [1]. In 2015, the World Health Organization recommended the use of oral pre-exposure prophylaxis (PrEP) by people at substantial risk of HIV acquisition [2], following randomized studies demonstrating significant reductions in HIV incidence among persons adhering to oral PrEP [3–6]. These guidelines were subsequently adopted by many African countries with high HIV burden, including Uganda [7]. However, population-based data on PrEP uptake in African populations outside of PEPFAR program data since PrEP guidelines were adopted are rare.

Beginning in 2017, the United States Centers for Disease Control through the President's Emergency Plan for AIDS Relief (PEFPAR) initiated PrEP implementation projects in selected African populations at high risk for HIV, including Lake Victoria fishing communities in eastern Africa. Fishing communities in the Lake Victoria basin have among the highest HIV incidence rates globally, with adult HIV seroprevalence typically exceeding 20% [8]. While these communities were among those prioritized for early rollout of oral PrEP, limited available programmatic and qualitative data suggests major challenges with retaining eligible individuals in these communities in PrEP programs [9,10]. For example, in a program evaluation among fisherfolk in southern Uganda, Kagaayi et al. reported >90% oral PrEP uptake among those screened and eligible, but median retention time among persons who initiated PrEP was only 45 days [10]. It remains unclear to what extent PrEP screening programs have reached individuals at substantial risk of HIV in Lake Victoria fishing communities and how this may have affected overall population-level patterns of PrEP use and retention.

The Rakai Community Cohort Study (RCCS) is an ongoing population HIV surveillance cohort in southern Uganda, including four Lake Victoria fishing communities with ~40% HIV seroprevalence [11]. The largest of these four fishing communities was the site of a PEPFAR-supported PrEP implementation project, which began in October 2017. Here, we used cross-sectional population-level data from this fishing community to assess levels and factors associated with PrEP awareness, ever use, and discontinuation following PrEP implementation in 2018. In contrast to prior programmatic and clinic-based assessments, we evaluated PrEP eligibility among

HIV-seronegative persons at a population-level and estimated the extent to which individuals at substantial risk of HIV and therefore PrEP-eligible were aware of and engaged with PrEP services. Understanding patterns of PrEP awareness, ever-use, and discontinuation at a population-level may inform future PrEP implementation strategies in Lake Victoria fisherfolk and other populations at substantial risk of HIV acquisition in Africa [12].

## Methods

### Study population and setting

This cross-sectional study was nested in the RCCS, an open population-based HIV census and surveillance cohort in south central Uganda, including four Lake Victoria fishing communities. The RCCS is conducted by the Rakai Health Sciences Program (RHSP), which is both an HIV research organization and implementer of PEPFAR-funded HIV prevention and treatment services. The RCCS has been described in detail elsewhere [10,11,13]. In brief, a detailed household census is conducted prior to each RCCS survey. The census enumerates all household members irrespective of age and is followed by a survey of age-eligible persons 15-49 years and resident in study communities for ≥6 months. The RCCS survey obtains data on participant demographics, sexual behaviors, recent sexual partnerships, HIV service utilization, including information on past and current PrEP use, among other data. Consenting survey participants are also tested for HIV using a validated rapid HIV testing algorithm [7] with confirmation by an enzyme immunoassay for persons testing HIV-seropositive for the first time. All RCCS participants are linked to HIV prevention services, as well as care and treatment if HIV-seropositive per Ugandan clinical guidelines.

Our analysis was restricted to HIV-seronegative adolescents and adults aged 15–49 years living in the largest of the four Lake Victoria fishing communities under RCCS surveillance. Data from this community were obtained between September 3, 2018 and December 19, 2018 as part of the nineteenth survey round of RCCS data collection. The analysis timeline was purposively selected to allow adequate time for PrEP implementation (~1 year) as well as to avoid possible confounding effect of COVID 19 pandemic, which began in March 2020 in Uganda. This particular fishing community is located along the banks of Lake Victoria in Kyotera district near Uganda's border with Tanzania, where the first HIV cases were identified in Eastern Africa [14].

### Ethics

This study was approved by the Research Ethics Committee of the Uganda Virus Research Institute (GC/127/08/12/137) and the Johns Hopkins School of Medicine Institutional Review Board (IRB00217467), and was registered with the Uganda National Council for Science and Technology (HS 540). All RCCS participants provide written informed consent (or assent with parental consent if < 18 years) prior to survey participation. This project was reviewed in accordance with CDC human research protection procedures and was determined to be research, but CDC investigators did not interact with human subjects or have access to identifiable data or specimens for research purposes.

### The oral PrEP programme

Oral PrEP (tenofovir disoproxil fumarate [TDF] and lamivudine [3TC]) was first rolled out among key and priority populations in southern Uganda, including among Lake Victoria fisherfolk in 2017. Details of its implementation have been described previously [10]. In the Lake Victoria fishing community in this study, residents were initially mobilized and sensitized about PrEP through RHSP-supported community outreach efforts. As part of community outreach, residents were referred to health facilities for PrEP eligibility screening, which was done using a risk-screening tool. Individuals were deemed PrEP-eligible if they reported at least one of the following risk factors: 1) vaginal sexual intercourse with more than one partner of unknown HIV status in the past six months; 2) vaginal sex without a condom in the past six months; 3) anal sexual intercourse in the past six months; 4) sex in exchange for money, goods or a service in the last six months; 5) injecting drugs in the past

six months; 6) diagnosis with a sexually transmitted infection more than once in the past twelve months; 7) post-exposure prophylaxis for sexual exposure to HIV in the past six months; and 8) having a sexual partner with HIV who was not on ART. Eligible persons who initiated PrEP were then followed on a quarterly basis at health facilities for adherence counseling, HIV retesting, and evaluation of side effects. In this current study, we assessed likely PrEP eligibility among RCCS participants by classifying them as having substantial HIV risk based on criteria previously described by Ssempijja et al [15]. Criteria were analogous, though somewhat different to the Uganda's national prep eligibility criteria above. Briefly, individuals were considered to have substantial HIV risk (i.e., likely PrEP-eligible) if they were HIV-seronegative and reported at least one of the following risk behaviors in the last year: having multiple sexual partners of unknown HIV serostatus, having genital ulcer disease, having non-marital sex without a condom, or engaging in transactional sex (considered to be sexual exploitation for those <18 years by the United Nations Convention on the Rights of the Child [16]).

### Primary outcomes

Our primary outcomes included PrEP awareness, PrEP ever use and PrEP discontinuation assessed by self-report at the time of RCCS survey. PrEP awareness and PrEP ever use were defined as self-reported "yes" responses to the questions: "Have you ever heard about a way to prevent HIV which involves an HIV-seronegative person taking a daily pill called PrEP to reduce their risk of acquiring HIV while in a sexual relationship with someone who might be HIV-positive?" and "Have you ever used PrEP?". PrEP discontinuation was defined as a self-reported response "no" to current PrEP use but "yes" to PrEP ever-use in the same survey. We ascertained reasons for PrEP discontinuation from a multiple response question that began by asking participants who had discontinued PrEP: "Why did you stop using PrEP?".

### Statistical analysis

We first described baseline demographic and behavioral characteristics of study participants by sex, with categorical variables reported as frequencies and percentages and continuous variables as medians with interquartile ranges (IQR). Next, we assessed the proportion self-reporting PrEP awareness and ever use by age and sex with proportions reported as percentages (i.e., prevalence). Individual-level correlates of PrEP awareness and ever use were evaluated for male and female participants separately using modified Poisson regression with robust variance estimators with and without adjustment for age. Associations were reported as unadjusted and age-adjusted prevalence ratios (PR) with 95% confidence intervals (95% CI). Correlates evaluated included age, educational level, occupation, recent in-migration to the community (since last RCCS survey; ~18 month interval), marital status, number of sexual partners in the past year, perceived HIV risk, intimate partner violence (i.e., self-reported experience of domestic and/or sexual abuse), substantial HIV risk/likely PrEP eligibility at time of survey, self-reported HIV test in the past year, use of at least one family planning method, and transactional sex with ≥1 of four most recent sexual partners in the past year. Given that PrEP use is driven by PrEP eligibility, we also conducted stratified analyses of PrEP ever use by substantial HIV risk/likely PrEP eligibility. We conducted similar analyses for PrEP discontinuation among those reporting having ever used PrEP; however, analyses were not stratified by sex due to limited sample size. Analyses were performed using Stata Version 17.

### Inclusivity in global research

Additional information regarding the ethical, cultural and scientific considerations specific to inclusivity in global research is included in the Supporting Information (S1 Checklist).

## Results

### Study population

There were 2,701 survey participants, of whom 1,401 (64.4%) tested HIV seronegative, including 577 (42.3%) women. Of these 1,401 participants, 1,363 (97%) reported having ever been tested for HIV and were subsequently asked about

their PrEP awareness and use. Median age among HIV seronegative participants who reported ever testing for HIV was 31 (IQR: 25-38) and 27 (IQR: 22-33) years among male and female participants, respectively (Table 1). The majority had at least some primary education, an HIV test within the past 12 months, and were married. Nearly half (47.7%) of all male participants and 29.3% of female participants were classified as being PrEP-eligible; however, female participants were more likely to report that they were very likely to acquire HIV (51.5% vs. 36.8%, respectively).

## Prevalence and correlates of PrEP awareness

Overall, 85.5% (n=1166/1363) of participants self-reported awareness of PrEP (Table 2). The proportion of participants with PrEP awareness did not significantly vary by sex (86.5% [680/786] among male vs. 84.2% [486/577] among female participants) but tended to be lower among adolescents 15-19 years relative to older age groups (Fig 1). Among male participants, awareness of PrEP was significantly lower among those who self-reported transactional sex in the last year, those who self-reported being not at risk or having unknown risk of HIV acquisition, and those who self-reported not having an HIV test in the past year (Table 2). Conversely, men who reported using a family planning method versus those who did not were more likely to be aware of PrEP, as were currently married men compared to men who had never been married. Similar to men, PrEP awareness was significantly lower among female participants with lower levels of HIV risk perception, but higher among those using family planning methods (Table 2). While PrEP awareness was significantly higher among women who reported multiple sexual partners and who were likely PrEP eligible, this was not the case for men (Table 2).

**Table 1. Characteristics of male and female study participants in the largest of four Lake Victoria fishing communities under RCCS surveillance in south central Uganda, between September and December of 2018.**

|  | Male, N (%) | Female, N (%) |
|---|---|---|
|  | 786 (57.7%) | 577 (42.3%) |
| Age, median (IQR) | 31 (25–38) | 27 (22–33) |
| Substantial HIV risk/likely PrEP eligible | 375 (47.7%) | 169 (29.3%) |
| Education level |  |  |
| None | 39 (5.0%) | 32 (5.6%) |
| Primary | 559 (71.1%) | 336 (58.2%) |
| Secondary/Tertiary | 188 (23.9%) | 209 (36.2%) |
| Marital status |  |  |
| Never married | 157 (20.0%) | 56 (9.7%) |
| Currently married | 440 (56.0%) | 390 (67.6%) |
| Previously married | 189 (24.1%) | 131 (22.7%) |
| Recent in-migrant* | 126 (16.0%) | 147 (25.5%) |
| HIV test within the last year | 580 (73.8%) | 486 (81.1%) |
| Using family planning** | 239 (30.4%) | 235 (45.7%) |
| Self-perceived HIV risk |  |  |
| Very likely | 297 (37.8%) | 297 (51.5%) |
| Somewhat likely | 289 (36.8%) | 171 (29.6%) |
| Unlikely | 109 (13.9%) | 51 (8.4%) |
| Not at all/don't know | 91 (11.6%) | 58 (10.1%) |

*Participant in-migrated to community since prior survey (~18-month survey interval between RCCS survey round 19 [current round] and RCCS survey round 18 [prior round]).

**Self-reported use of at least one family planning method at the time of the survey

Table 2. Individual-level correlates of PrEP awareness among male and female RCCS participants in a Lake Victoria Fishing Community in south central Uganda in 2019.

| Characteristics | Male participants (n=786) | | | | | Female participants (n=577) | | | | |
|---|---|---|---|---|---|---|---|---|---|---|
| | No. Reporting/Total (%) | Unadjusted PR (95% CI) | P-value | Age-adjusted PR (95%CI) | P-value | No. Reporting/Total (%) | Unadjusted PR (95% CI) | P-value | Age-adjusted PR (95%CI) | P-value |
| **Age group (years)** | | | | | | | | | | |
| 15-19 | 42/57 (73.7) | 0.85 (0.72–1.00) | 0.055 | 0.85 (0.72–1.00) | 0.055 | 48/69 (69.6) | **0.80 (0.67–0.94)** | **0.007** | **0.80 (0.67–0.94)** | **0.007** |
| 20-24 | 113/130 (86.9) | Ref | Ref | Ref | Ref | 133/152 (87.5) | Ref | Ref | Ref | Ref |
| 25-29 | 148/166 (89.2) | 1.03 (0.94–1.12) | 0.560 | 1.03 (0.94–1.12) | 0.560 | 110/131 (84) | 0.96 (0.87–1.06) | 0.401 | 0.96 (0.87–1.06) | 0.401 |
| 30-34 | 131/155 (84.5) | 0.97 (0.88–1.07) | 0.562 | 0.97 (0.88–1.07) | 0.562 | 82/94 (87.2) | 1.00 (0.90–1.10) | 0.951 | 1.00 (0.90–1.10) | 0.951 |
| 35-39 | 117/130 (90.0) | 1.04 (0.95–1.13) | 0.438 | 1.04 (0.95–1.13) | 0.438 | 54/59 (91.5) | 1.05 (0.95–1.15) | 0.370 | 1.05 (0.95–1.15) | 0.370 |
| 40-44 | 83/92 (90.2) | 1.04 (0.94–1.14) | 0.442 | 1.04 (0.94–1.14) | 0.442 | 39/47 (83) | 0.95 (0.82–1.09) | 0.467 | 0.95 (0.82–1.09) | 0.467 |
| 45-49 | 46/56 (82.1) | 0.95 (0.82–1.09) | 0.426 | 0.95 (0.82–1.09) | 0.426 | 20/25 (80) | 0.91 (0.74–1.12) | 0.392 | 0.91 (0.74–1.12) | 0.392 |
| **Educational status** | | | | | | | | | | |
| None | 32/39 (82.1) | 0.93 (0.80–1.08) | 0.359 | 0.91 (0.78–1.06) | 0.232 | 26/32 (81.3) | 0.97 (0.81–1.15) | 0.713 | 0.96 (0.81–1.13) | 0.598 |
| Primary | 492/559 (88.0) | Ref | Ref | Ref | Ref | 282/336 (83.9) | Ref | Ref | Ref | Ref |
| Secondary/ Tertiary | 156/188 (83.0) | 0.94 (0.88–1.01) | 0.107 | 0.94 (0.88–1.01) | 0.107 | 178/209 (85.2) | 1.01 (0.94–1.09) | 0.696 | 1.02 (0.94–1.10) | 0.676 |
| **Primary occupation** | | | | | | | | | | |
| Other | 289/344 (84.0) | 0.94 (0.89–1.00) | 0.035 | 0.95 (0.90–1.00) | 0.072 | 331/390 (84.9) | 1.03 (0.94–1.13) | 0.504 | 1.03 (0.94–1.12) | 0.553 |
| Agrarian | 38/47 (80.9) | 0.90 (0.78–1.04) | 0.171 | 0.91 (0.78–1.06) | 0.217 | 43/51 (84.3) | 1.02 (0.89–1.18) | 0.745 | 1.00 (0.87–1.16) | 0.967 |
| Fishing | 353/395 (89.4) | Ref | Ref | Ref | Ref | – | – | – | – | – |
| Housework | – | – | – | – | – | 112/136 (82.4) | Ref | Ref | Ref | Ref |
| **Recent in-migrant\*** | | | | | | | | | | |
| No | 579/660 (87.7) | Ref | Ref | Ref | Ref | 372/430 (86.5) | Ref | Ref | Ref | Ref |
| Yes | 101/126 (80.2) | 0.91 (0.83–1.00) | 0.053 | 0.92 (0.84–1.01) | 0.073 | 114/147 (77.6) | **0.90 (0.82–0.99)** | **0.024** | 0.91 (0.83–1.00) | 0.062 |
| **Marital status** | | | | | | | | | | |
| Never married | 124/157 (79.0) | Ref | Ref | Ref | Ref | 40/56 (71.4) | Ref | Ref | Ref | Ref |
| Currently married | 394/440 (89.5) | **1.13 (1.04–1.24)** | **0.005** | **1.11 (1.01–1.23)** | **0.037** | 328/390 (84.1) | 1.18 (0.99–1.40) | 0.062 | 1.08 (0.92–1.27) | 0.331 |
| Previously married | 162/189 (85.7) | 1.09 (0.98–1.20) | 0.107 | 1.06 (0.95–1.19) | 0.295 | 118/131 (90.1) | 1.26 (1.06–1.50) | 0.009 | 1.16 (0.99–1.37) | 0.067 |
| **Number of sexual partners in last year** | | | | | | | | | | |
| 0 or 1 | 289/338 (85.5) | Ref | Ref | Ref | Ref | 387/467 (82.9) | Ref | Ref | Ref | Ref |
| 2 | 165/184 (89.7) | 1.05 (0.98–1.12) | 0.156 | 1.04 (0.98–1.12) | 0.194 | 71/80 (88.8) | 1.07 (0.98–1.17) | 0.128 | 1.08 (0.99–1.18) | 0.086 |
| >=3 | 226/264 (85.6) | 1.00 (0.94–1.07) | 0.972 | 0.98 (0.92–1.05) | 0.617 | 28/30 (93.3) | **1.13 (1.01–1.25)** | **0.025** | **1.13 (1.02–1.25)** | **0.023** |
| **Perceived HIV risk** | | | | | | | | | | |
| Very likely | 260/297 (87.5) | Ref | Ref | Ref | Ref | 258/297 (86.9) | Ref | Ref | Ref | Ref |
| Somewhat likely | 256/289 (88.6) | 1.01 (0.95–1.07) | 0.698 | 1.02 (0.96–1.09) | 0.488 | 152/171 (88.9) | 1.02 (0.95–1.10) | 0.514 | 1.02 (0.95–1.10) | 0.512 |

*(Continued)*

**Table 2.** (Continued)

| Characteristics | Male participants (n=786) | | | | | Female participants (n=577) | | | | |
|---|---|---|---|---|---|---|---|---|---|---|
| | No. Reporting/Total (%) | Unadjusted PR (95% CI) | P-value | Age-adjusted PR (95%CI) | P-value | No. Reporting/Total (%) | Unadjusted PR (95% CI) | P-value | Age-adjusted PR (95%CI) | P-value |
| Unlikely | 98/109 (89.9) | 1.03 (0.95–1.11) | 0.493 | 1.04 (0.97–1.12) | 0.290 | 36/51 (70.6) | **0.81 (0.68–0.98)** | **0.026** | **0.82 (0.68–0.99)** | **0.035** |
| Not at all/Don't know | 66/91 (72.5) | **0.83 (0.72–0.95)** | **0.006** | **0.86 (0.75–0.98)** | **0.026** | 40/58 (69.0) | **0.79 (0.66–0.95)** | **0.011** | **0.83 (0.70–0.99)** | **0.038** |
| **Intimate partner violence** | | | | | | | | | | |
| No | 497/574 (86.6) | Ref | Ref | Ref | Ref | 316/375 (84.3) | Ref | Ref | Ref | Ref |
| Yes | 151/168 (89.9) | 1.04 (0.98–1.10) | 0.223 | 1.03 (0.97–1.10) | 0.270 | 141/158 (89.2) | 1.06 (0.99–1.14) | 0.107 | 1.07 (1.00–1.15) | 0.068 |
| **Substantial HIV risk/likely PrEP eligible** | | | | | | | | | | |
| No | 356/411 (86.6) | Ref | Ref | Ref | Ref | 338/408 (82.8) | Ref | Ref | Ref | Ref |
| Yes | 324/375 (86.4) | 1.00 (0.94–1.05) | 0.929 | 0.99 (0.94–1.05) | 0.828 | 148/169 (87.6) | 1.06 (0.98–1.14) | 0.131 | **1.08 (1.00–1.16)** | **0.044** |
| **Most recent HIV test** | | | | | | | | | | |
| < 1 year | 514/580 (88.6) | Ref | Ref | Ref | Ref | 399/468 (85.3) | Ref | Ref | Ref | Ref |
| >=1 year | 166/206 (80.6) | **0.91 (0.85–0.98)** | **0.011** | **0.91 (0.85–0.98)** | **0.012** | 87/109 (79.8) | 0.94 (0.85–1.04) | 0.204 | 0.93 (0.84–1.03) | 0.176 |
| **Current FP use** | | | | | | | | | | |
| No | 463/547 (84.6) | Ref | Ref | Ref | Ref | 219/279 (78.5) | Ref | Ref | Ref | Ref |
| Yes | 217/239 (90.8) | **1.07 (1.02–1.13)** | **0.011** | **1.06 (1.01–1.12)** | **0.030** | 213/235 (90.6) | **1.15 (1.07–1.24)** | **<0.001** | **1.15 (1.06–1.23)** | **<0.001** |
| **Transactional sex in last year** | | | | | | | | | | |
| No | 503/566 (88.9) | Ref | Ref | Ref | Ref | 387/453 (85.4) | Ref | Ref | Ref | Ref |
| Yes | 152/187 (81.3) | **0.91 (0.85–0.99)** | **0.019** | **0.91 (0.84–0.98)** | **0.009** | 75/87 (86.2) | 1.01 (0.92–1.11) | 0.848 | 1.02 (0.93–1.12) | 0.671 |

PR = prevalence ratio; CI = confidence interval; FP = family planning;

*Participant in-migrated to community since prior survey (~18-month survey interval between RCCS survey round 19 [current round] and RCCS survey round 18 [prior round]),

**Self-reported use of at least one family planning method at the time of the survey,

***Sexual exploitation for respondents <18 years of age.

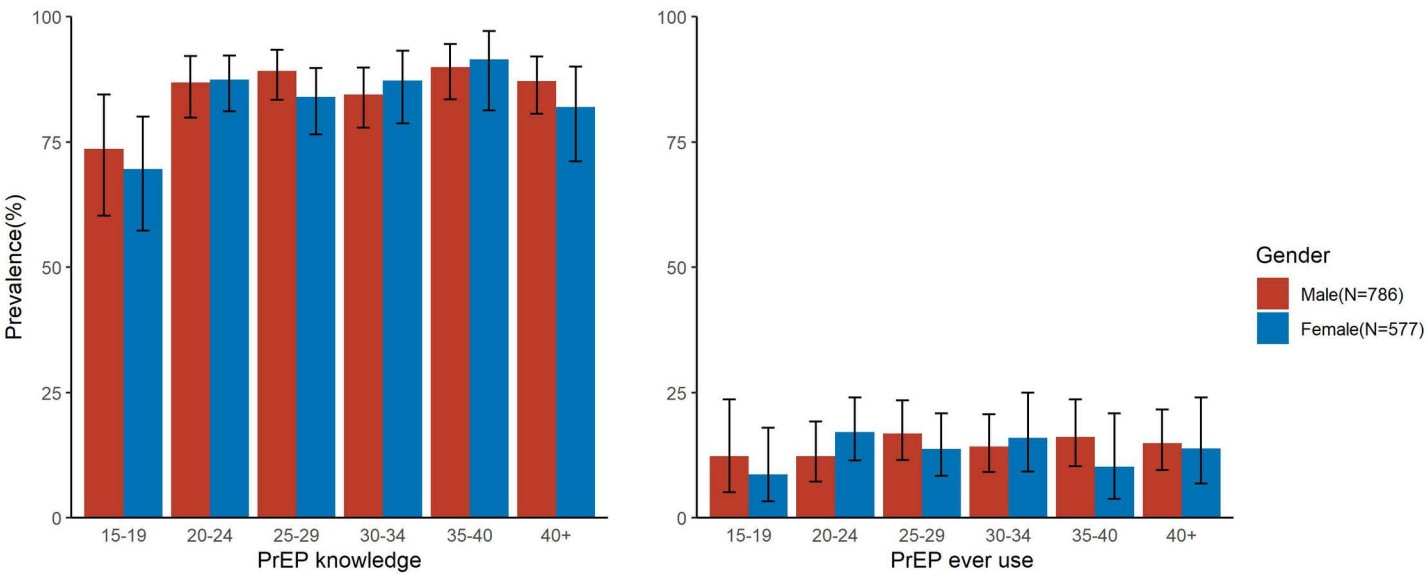

**Fig 1. Prevalence of self-reported PrEP awareness and ever use by sex and age among RCCS participants in a Lake Victoria Fishing community in south central Uganda in 2019.**

## Prevalence and correlates of PrEP ever use

Prevalence of PrEP ever use was 14.5% (n=197/1363) and was generally similar between male and female participants (14.8% [116/786] versus 14% [81/577], respectively), but similar to PrEP awareness, lower among adolescents (Fig 2). Participants classified as being at substantial HIV risk (i.e., likely PrEP eligible) at time of survey were significantly more

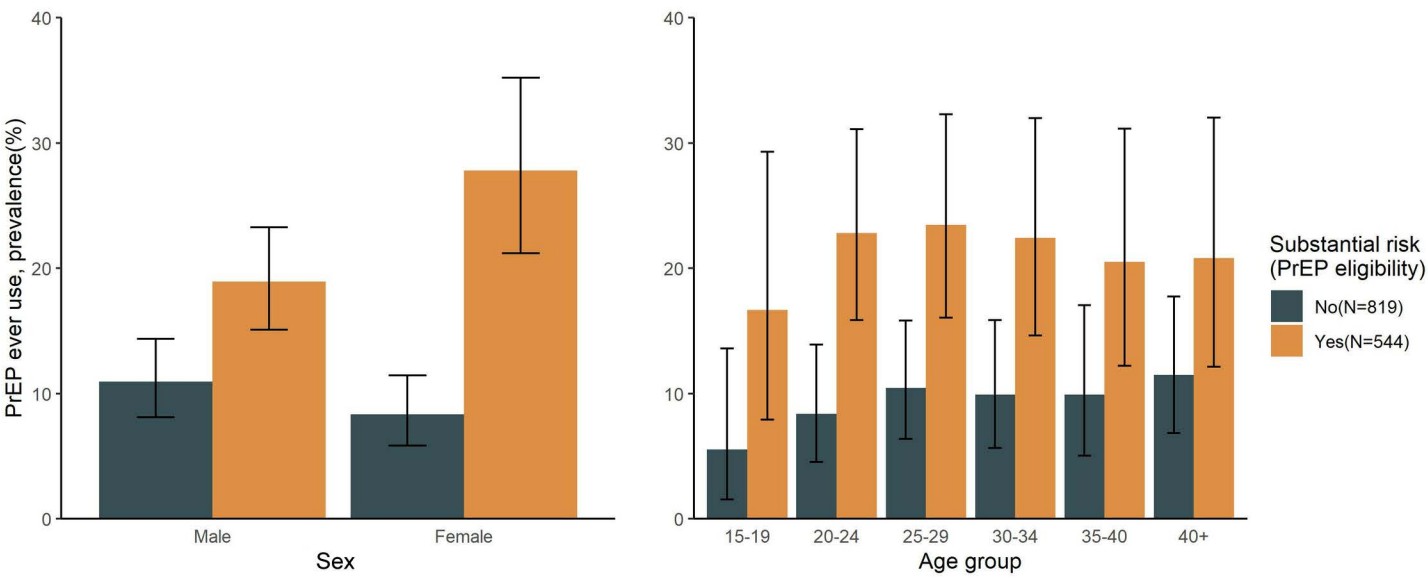

**Fig 2. Prevalence of self-reported PrEP ever use among RCSS participants with and without substantial HIV risk (i.e., likely PrEP eligibility) at time of survey in a Lake Victoria Fishing community in south central Uganda in 2019.**

likely to report having used PrEP compared to those who were not at substantial HIV risk (21.7% [n=118/544] vs 9.7% [79/819]; adjPR= 2.27; 95%CI: 1.75 – 2.95; p= <0.001). While proportionately more male than female participants were at substantial HIV risk and likely PrEP eligible (47.7% [n=375/786] versus 29.3% [n=169/577]; S1 Table), prevalence of PrEP ever use was significantly higher among female versus male participants likely PrEP eligible (27.8% [47/169] versus 18.9% [71/375]; PRR=1.47; 95%CI: 1.07 – 2.02; p= 0.019) (Fig 2).

Among male participants, having three or more sexual partners in the past year, higher HIV risk perception, or having an HIV test in the last year was significantly associated with higher levels of PrEP ever use (Table 3). This was also the case among female participants; however, girls and women were also significantly more likely to report having used PrEP if they reported intimate partner violence or transactional sex/sexual exploitation (Table 3). Female participants who reported using a family planning method also tended to have higher levels of PrEP ever use as compared to female participants who did not report using a family planning method (adjPR=1.58; 95%CI:0.99-2.52). Individual-level correlates of PrEP ever use were generally similar among participants at substantial HIV risk (S1 Table).

### Prevalence and correlates of PrEP discontinuation

Among participants who reported ever using PrEP (n=197), 47.7% (n= 94) reported that they were not currently using PrEP at time of survey (i.e., discontinued PrEP). The most common reasons for reporting PrEP discontinuation were: side effects (39.4%; 37/94), pill burden (20.2%; 19/94), low risk perception (12.8%; 12/94), trouble transferring between clinics (9.6%; 9/94), not ready to take PrEP (7.5%; 7/94); and lack of pills (5.3%; 5/94). Other less common reasons included stigma (n=3), use of other HIV prevention methods (n=4), skepticism about the efficacy of PrEP (n=2), and long distance to the clinic (n=2). With the exception of higher educational status, which was associated with higher levels of discontinuation, levels of PrEP discontinuation did not substantially vary by age or any other individual- level factors, including being at substantial HIV risk (S2 Table).

### Discussion

In this cross-sectional, population-based study of oral PrEP use in a Ugandan Lake Victoria fishing community following a community-based PrEP implementation project, we observed very high levels of PrEP awareness but low levels of PrEP use. PrEP awareness and ever use tended to be lower among adolescents, while PrEP ever use was higher among those with elevated HIV risk perception, and persons reporting having had an HIV test in the last year. While male participants had higher levels of HIV-associated risk behaviors, they had significantly lower levels of PrEP use compared to their female counterparts. Among those who reported having ever used PrEP, less than 50% were currently using PrEP at time of survey, and most people at substantial HIV risk and likely PrEP-eligible had never used PrEP.

More than 80% of participants in our study population were aware of PrEP. Although few prior studies have assessed PrEP awareness at a community-level, our findings are consistent with a 2019 population-based survey of persons aged 18-24 in six counties in western Kenya [17], which similarly found that 84% of persons were aware of PrEP. However, studies of high-risk key populations in Uganda and other African countries have reported substantially lower levels of PrEP awareness, with typically less than half of participants being PrEP-aware [18–20]. Levels of PrEP awareness in this study were likely high because of intensive community-based outreach done as part of the implementation project. Such activities have been previously shown to improve PrEP awareness and uptake in high burden areas such as Lake Victoria fishing communities [21].

While PrEP awareness was generally high, both awareness and use were lowest among adolescent boys and girls aged 15-19 years. Such relatively low levels of PrEP use among adolescents compared to other age groups also have been observed in other settings [22,23]. Despite low PrEP use, we found that participants under 25 years of age had the highest prevalence of substantial HIV risk behaviors, and across Africa, adolescent girls are typically the age-group most at risk of HIV acquisition [24]. Thus, our findings suggest that innovative approaches may be needed to improve the PrEP

Table 3. Individual-level correlates of PrEP ever use among female and male RCCS participants in a Lake Victoria fishing community in south central Uganda in 2019.

| Characteristics | Male participants (n=786) | | | | | Female participants (n=577) | | | | |
|---|---|---|---|---|---|---|---|---|---|---|
| | No. Reporting/Total (%) | Unadjusted PR (95%CI) | P-value | Age-adjusted PR (95%CI) | P-value | No. Reporting/Total (%) | Unadjusted PR (95%CI) | P-value | Age-adjusted PR (95%CI) | P-value |
| **Age group (years)** | | | | | | | | | | |
| 15-19 | 7/57 (12.3) | 1.00 (0.43–2.29) | 0.996 | 1.00 (0.43–2.29) | 0.996 | 6/69 (8.7) | 0.51 (0.22–1.18) | 0.115 | 0.51 (0.22–1.18) | 0.115 |
| 20-24 | 16/130 (12.3) | Ref | Ref | Ref | Ref | 26/152 (17.1) | Ref | Ref | Ref | Ref |
| 25-29 | 28/166 (16.9) | 1.37 (0.77–2.42) | 0.279 | 1.37 (0.77–2.42) | 0.279 | 18/131 (13.7) | 0.80 (0.46–1.40) | 0.439 | 0.80 (0.46–1.40) | 0.439 |
| 30-34 | 22/155 (14.2) | 1.15 (0.63–2.10) | 0.642 | 1.15 (0.63–2.10) | 0.642 | 15/94 (16.0) | 0.93 (0.52–1.67) | 0.815 | 0.93 (0.52–1.67) | 0.815 |
| 35-39 | 21/130 (16.2) | 1.31 (0.72–2.40) | 0.377 | 1.31 (0.72–2.40) | 0.377 | 6/59 (10.2) | 0.59 (0.26–1.37) | 0.223 | 0.59 (0.26–1.37) | 0.223 |
| 40-44 | 19/92 (20.7) | 1.68 (0.91–3.09) | 0.096 | 1.68 (0.91–3.09) | 0.096 | 7/47 (14.9) | 0.87 (0.40–1.88) | 0.724 | 0.87 (0.40–1.88) | 0.724 |
| 45-49 | 3/56 (5.4) | 0.44 (0.13–1.44) | 0.172 | 0.44 (0.13–1.44) | 0.172 | 3/25 (12.0) | 0.70 (0.23–2.15) | 0.535 | 0.70 (0.23–2.15) | 0.535 |
| **Educational Status** | | | | | | | | | | |
| None | 9/39 (23.1) | 1.59 (0.87–2.92) | 0.133 | 1.45 (0.79–2.65) | 0.230 | 6/32 (18.8) | 1.31 (0.61–2.83) | 0.488 | 1.42 (0.64–3.13) | 0.386 |
| Primary | 81/559 (14.5) | Ref | Ref | Ref | Ref | 48/336 (14.3) | Ref | Ref | Ref | Ref |
| Secondary/Tertiary | 26/188 (13.8) | 0.95 (0.63–1.44) | 0.824 | 0.97 (0.64–1.46) | 0.878 | 27/209 (12.9) | 0.90 (0.58–1.40) | 0.653 | 0.90 (0.57–1.41) | 0.633 |
| **Primary occupation** | | | | | | | | | | |
| Other | 43/344 (12.5) | 0.71 (0.50–1.00) | 0.052 | 0.72 (0.50–1.03) | 0.074 | 57/390 (14.6) | 1.42 (0.82–2.46) | 0.213 | 1.51 (0.86–2.66) | 0.147 |
| Agrarian | 3/47 (6.4) | 0.36 (0.12–1.10) | 0.073 | 0.36 (0.11–1.11) | 0.075 | 10/51 (19.6) | 1.90 (0.90–4.01) | 0.090 | 2.06 (0.96–4.41) | 0.063 |
| Fishing | 70/395 (17.7) | Ref | Ref | Ref | | | | | | |
| Housework | | | | | | 14/136 (10.3) | Ref | Ref | Ref | Ref |
| **Recent in-migrant*** | | | | | | | | | | |
| No | 93/660 (14.1) | Ref | Ref | Ref | Ref | 62/430 (14.4) | Ref | Ref | Ref | Ref |
| Yes | 23/126 (18.3) | 1.30 (0.86–1.96) | 0.222 | 1.38 (0.91–2.09) | 0.133 | 19/147 (12.9) | 0.90 (0.56–1.45) | 0.655 | 0.89 (0.54–1.46) | 0.653 |
| **Marital status** | | | | | | | | | | |
| Never married | 19/157 (12.1) | Ref | Ref | Ref | Ref | 5/56 (8.9) | Ref | Ref | Ref | Ref |
| Currently married | 57/440 (13.0) | 1.07 (0.66–1.74) | 0.784 | 0.98 (0.51–1.88) | 0.953 | 52/390 (13.3) | 1.49 (0.62–3.58) | 0.369 | 1.29 (0.49–3.36) | 0.608 |
| Previously married | 40/189 (21.2) | 1.75 (1.06–2.89) | **0.030** | 1.59 (0.83–3.05) | 0.160 | 24/131 (18.3) | 2.05 (0.82–5.11) | 0.122 | 1.86 (0.67–5.16) | 0.231 |
| **Number of sexual partners in last year** | | | | | | | | | | |
| 0 or 1 | 38/338 (11.2) | Ref | Ref | Ref | Ref | 50/467 (10.7) | Ref | Ref | Ref | Ref |
| 2 | 28/184 (15.2) | 1.35 (0.86–2.13) | 0.191 | 1.36 (0.86–2.15) | 0.187 | 18/80 (22.5) | **2.10 (1.29–3.41)** | **0.003** | **2.25 (1.39–3.66)** | **0.001** |
| >=3 | 50/264 (18.9) | **1.68 (1.14–2.49)** | **0.009** | **1.68 (1.13–2.51)** | **0.011** | 13/30 (43.3) | **4.05 (2.49–6.58)** | **<0.001** | **4.22 (2.56–6.96)** | **<0.001** |
| **Perceived HIV risk** | | | | | | | | | | |
| Very likely | 60/297 (20.2) | Ref | Ref | Ref | Ref | 58/297 (19.5) | Ref | Ref | Ref | Ref |
| Somewhat likely | 36/289 (12.5) | **0.62 (0.42–0.90)** | **0.013** | **0.64 (0.43–0.93)** | **0.021** | 19/171 (11.1) | **0.57 (0.35–0.92)** | **0.022** | **0.55 (0.34–0.90)** | **0.017** |

*(Continued)*

Table 3.  (Continued)

| Characteris-tics | Male participants (n=786) | | | | | Female participants (n=577) | | | | |
|---|---|---|---|---|---|---|---|---|---|---|
| | No. Reporting/ Total (%) | Unadjusted PR (95%CI) | P-value | Age-adjusted PR (95%CI) | P-value | No. Reporting/ Total (%) | Unadjusted PR (95%CI) | P-value | Age-adjusted PR (95%CI) | P-value |
| Unlikely | 12/109 (11.0) | 0.54 (0.31–0.97) | 0.040 | 0.57 (0.32–1.00) | 0.052 | 2/51 (3.9) | 0.20 (0.05–0.80) | 0.023 | 0.20 (0.05–0.79) | 0.022 |
| Not at all/ Don't know | 8/91 (8.8) | 0.44 (0.22–0.88) | 0.020 | 0.46 (0.21–0.99) | 0.046 | 2/58 (3.5) | 1.18 (0.04–0.70) | 0.014 | 0.18 (0.04–0.73) | 0.017 |
| **Intimate partner violence** | | | | | | | | | | |
| No | 85/574 (14.8) | Ref | Ref | Ref | Ref | 50/375 (13.3) | Ref | Ref | Ref | Ref |
| Yes | 30/168 (17.9) | 1.21 (0.82–1.76) | 0.333 | 1.19 (0.82–1.74) | 0.367 | 31/158 (19.6) | 1.47 (0.98–2.21) | 0.064 | 1.52 (1.01–2.29) | 0.044 |
| **Substantial HIV risk/likely PrEP eligible** | | | | | | | | | | |
| No | 45/411 (11.0) | Ref | Ref | Ref | Ref | 34/408 (8.3) | Ref | Ref | Ref | Ref |
| Yes | 71/375 (18.9) | 1.73 (1.22–2.45) | 0.002 | 1.76 (1.24–2.49) | 0.002 | 47/169 (27.8) | 3.34 (2.23–5.00) | <0.001 | 3.45 (2.31–5.13) | <0.001 |
| **Most recent HIV test** | | | | | | | | | | |
| < 1 year | 103/580 (17.8) | Ref | Ref | Ref | Ref | 78/468 (16.7) | Ref | Ref | Ref | Ref |
| >=1 year | 13/206 (6.3) | 0.36 (0.20–0.62) | <0.001 | 0.36 (0.21–0.64) | <0.001 | 3/109 (2.8) | 0.17 (0.05–0.51) | 0.002 | 0.17 (0.05–0.52) | 0.002 |
| **Current FP use**** | | | | | | | | | | |
| No | 79/547 (14.4) | Ref | Ref | Ref | Ref | 30/279 (10.8) | Ref | Ref | Ref | Ref |
| Yes | 37/239 (15.5) | 1.07 (0.75–1.54) | 0.705 | 1.03 (0.71–1.48) | 0.885 | 41/235 (17.5) | 1.62 (1.05–2.51) | 0.030 | 1.58 (0.99–2.52) | 0.054 |
| **Transactional sex in last year**** | | | | | | | | | | |
| No | 87/566 (15.4) | Ref | Ref | Ref | Ref | 49/453 (10.8) | Ref | Ref | Ref | Ref |
| Yes | 29/187 (15.5) | 1.01 (0.69–1.49) | 0.964 | 1.00 (0.69–1.47) | 0.980 | 32/87 (36.8) | 3.40 (2.32–4.98) | <0.001 | 3.56 (2.43–5.22) | <0.001 |

PR = prevalence ratio; CI = confidence interval; FP = family planning;

*Participant in-migrated to community since prior survey (~18-month survey interval between RCCS survey round 19 [current round] and RCCS survey round 18 [prior round]);

**Self-reported use of at least one family planning method at the time of the survey,

***Sexual exploitation for respondents under 18 years of age.

care continuum among adolescents. Future research should identify barriers to PrEP awareness and use in this population to inform tailored outreach strategies.

Consistent with earlier studies in African populations at high risk for HIV [18,25,26], we found generally low levels of PrEP use despite high levels of PrEP awareness. Critically, the majority of participants at substantial HIV risk and likely PrEP eligible had never used PrEP. While men were more likely to be at substantial HIV risk than women, men were significantly less likely to have ever used PrEP compared to their female counterparts with similarly high levels of HIV-related risk factors. Notably, most men in our study were Lake Victoria fishermen, who are often highly mobile [27]. Prior studies have linked higher levels of mobility to reduced PrEP uptake and retention [9]. While most prior PrEP implementation studies in Africa have focused on girls and women, our findings underscore the need to tailor service delivery for men. Differentiated PrEP delivery models that include out-of-clinic options, event-driven PrEP, multi-month dispensing, and long-acting PrEP options (such as cabotegravir) could improve PrEP uptake among men, adolescents, and other mobile groups in Lake Victoria fishing communities [28,29].

We also observed that higher self-perceived HIV risk was associated with a higher propensity for PrEP ever-use irrespective of sex. This finding is consistent with those from prior studies, which have similarly reported that HIV risk perception underpins PrEP use in African populations [22,30,31]. We also found that PrEP use was strongly linked to recent HIV testing behavior. While this could be the result of HIV testing requirements for those who initiate and continue PrEP, an earlier study also conducted among Ugandan Lake Victoria fisherfolk observed that having tested for HIV within the last 6 months was associated with higher odds of willingness to use PrEP [32]. Moreover, participants in a qualitative study conducted among adolescents in the Western Cape province of South Africa expressed that the availability of the PrEP option to protect themselves against HIV acquisition motivates them to be aware of their HIV status and therefore test regularly [33]. In addition to recent HIV testing, those reporting using family planning also tended to have higher levels of PrEP awareness and PrEP ever-use. People using health services such as HIV testing and family planning may have more frequent contacts with the health care system, and thus, additional opportunities for exposure to PrEP. Better integration of HIV and sexual and reproductive services could improve PrEP use and other health outcomes [34,35].

Nearly half of the participants who had ever used PrEP in this study were no longer taking PrEP at time of survey. Although we were unable to specify the timing of PrEP initiation or discontinuation, our findings are congruent with those from a meta-analysis of longitudinal studies conducted in sub-Saharan Africa that found nearly half of individuals who initiated PrEP discontinued PrEP within 6 months [36]. It is possible these high levels of PrEP discontinuation in African populations are a result of dynamic HIV risk within individuals. For example, a recent population-based study, nested in the same cohort as this study in Uganda, reported that HIV risk waxes and wanes over time for many people [37]. Prior qualitative studies in Uganda also have linked PrEP discontinuation to stigma, PrEP side effects, and transportation barriers, which we similarly observed in this study [9,32]. In our study, the most common specified barriers were side effects and pill burden, rather than a change in self-perceived risk. Enhanced counseling for side effects, including that they wane over time, and expansion of event-driven PrEP to men, as well as prioritizing implementation of new PrEP agents including long acting cabotegravir and the dapivirine vaginal ring [38,39], may help attenuate these common reasons for discontinuation.

Our analysis has important limitations. First, this study was a cross-sectional analysis, so our ability to infer temporal relationships between demographic, behavioral, and health utilization factors with PrEP outcomes was limited. For example, we do not know whether high levels of HIV testing among those who had reported PrEP ever use were a cause or consequence of PrEP initiation. We also do not know how changes in PrEP eligibility within individuals may have changed over time and impacted PrEP outcomes, particularly PrEP discontinuation. Second, most data in this study were self-reported, including primary study outcomes, and are likely subject to measurement bias. Third, our assessment of PrEP eligibility from self-reported HIV risk factors in the RCCS only approximated those used in Uganda's national PrEP eligibility tool. Specifically, we considered only a subset of risk behaviors, and these were measured over the last year rather

than the prior 6 months. Fourth, we did not assess timing of PrEP discontinuation, and because only few people had ever reported PrEP use, we had limited power to detect factors associated with this outcome. Fifth, this study was conducted in only one Lake Victoria fishing community in Uganda with extremely high HIV burden, so results may not be generalizable to other populations. Lastly, data for these analyses were collected in 2018. While this allowed us to avoid the confounding influence of COVID-19 lockdowns on PrEP programs, factors associated with PrEP use may have changed in the intervening years. For example, PrEP scale-up in Uganda has since expanded.

In conclusion, this Lake Victoria fishing community with extremely high HIV prevalence had low levels of PrEP use despite high levels of PrEP awareness and PrEP eligibility. At the population level, PrEP use is linked to higher perceived HIV risk, HIV-associated risk factors, and recent HIV testing. Efforts that enhance awareness of HIV risk and increase linkage to PrEP through integrated health services may help increase PrEP use among HIV-seronegative persons in African settings with high HIV burden. Newer PrEP options that are long-acting may also be more suitable for mobile populations. More implementation research is needed to improve PrEP uptake and persistence among eligible people.

## Supporting information

**S1 Checklist.** Inclusivity in global research questionnaire.
(DOCX)

**S1 Table.** Individual-level correlates of PrEP ever use among 544 participants in a Lake Victoria Fishing community in southcentral Uganda in 2019 at substantial HIV risk/likely PrEP eligible stratified by sex.
(DOCX)

**S2 Table.** Individual-level correlates of PrEP discontinuation among female and male RCCS participants reporting PrEP ever use (n = 197) in a Lake Victoria fishing community in southcentral Uganda in 2019.
(DOCX)

## Author contributions

**Conceptualization:** Kauthrah Ntabadde, John Santelli, Larry William Chang, Gertrude Nakigozi, Mary Kate Grabowski.

**Data curation:** Robert Ssekubugu, Joseph Ssekasanvu.

**Formal analysis:** Kauthrah Ntabadde, Victor Ssempijja, Xinyi Feng.

**Funding acquisition:** Joseph Kagaayi, Larry William Chang, Mary Kate Grabowski.

**Investigation:** Joseph Kagaayi, Victor Ssempijja, Robert Ssekubugu, Ping Teresa Yeh, Aaron A R Tobian, Caitlin E. Kennedy, Lisa A. Mills, Stella Alamo, Philip Kreniske, John Santelli, Lisa J. Nelson, Steven J. Reynolds, Larry William Chang, Gertrude Nakigozi, Mary Kate Grabowski.

**Methodology:** Kauthrah Ntabadde, Victor Ssempijja, Mary Kate Grabowski.

**Project administration:** Robert Kairania, Joseph Lubwama, Robert Ssekubugu.

**Writing – original draft:** Kauthrah Ntabadde, Ping Teresa Yeh.

**Writing – review & editing:** Kauthrah Ntabadde, Joseph Kagaayi, Victor Ssempijja, Robert Ssekubugu, Ping Teresa Yeh, Aaron A R Tobian, Caitlin E. Kennedy, Lisa A. Mills, Stella Alamo, Philip Kreniske, John Santelli, Lisa J. Nelson, Steven J. Reynolds, Larry William Chang, Gertrude Nakigozi, Mary Kate Grabowski.

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
