## [Decision Letter · Decision Letter 0]

11 Oct 2024

PGPH-D-24-01886

Pre-exposure prophylaxis (PrEP) knowledge, use, and discontinuation among Lake Victoria fisherfolk in Uganda: a cross- sectional population-based study

Dear Kate,

Thank you for submitting your manuscript to PLOS Global Public Health. After careful consideration, we feel that it has merit but does not fully meet PLOS Global Public Health’s publication criteria as it currently stands. Therefore, we invite you to submit a revised version of the manuscript that addresses the points raised during the review process.

We look forward to receiving your revised manuscript.

Kind regards,

Collins Otieno Asweto, PhD

Academic Editor

Journal Requirements:

2. Your current Financial Disclosure states, “Bill and Melinda Gates Foundation (OPP1175094)”. However, it is missing in your funding information on the submission form. Please indicate by return email the full and correct funding information for your study and confirm the order in which funding contributions should appear. Please be sure to indicate whether the funders played any role in the study design, data collection and analysis, decision to publish, or preparation of the manuscript.

3. Uploaded as supplementary information.

4. Please provide separate figure files in .tif or .eps format.

5. We have noticed that you have uploaded Supporting Information files, but you have not included a list of legends. Please add a full list of legends for your Supporting Information files after the references list. 

Reviewers' comments:

Reviewer's Responses to Questions

**Comments to the Author**

1. Does this manuscript meet PLOS Global Public Health’s publication criteria ? Is the manuscript technically sound, and do the data support the conclusions? The manuscript must describe methodologically and ethically rigorous research with conclusions that are appropriately drawn based on the data presented.

Reviewer #1: Yes

Reviewer #2: Yes

2. Has the statistical analysis been performed appropriately and rigorously?

Reviewer #1: I don't know

Reviewer #2: Yes

3. Have the authors made all data underlying the findings in their manuscript fully available (please refer to the Data Availability Statement at the start of the manuscript PDF file)?

Reviewer #1: Yes

Reviewer #2: Yes

4. Is the manuscript presented in an intelligible fashion and written in standard English?

Reviewer #1: No

Reviewer #2: Yes

5. Review Comments to the Author

Reviewer #1: The authors explored an important topic of Public health importance. The findings will be valuable in implementing HIV preventions interventions. The manuscript can be improved by addressing the following issues;

The tense used in the manuscript should be uniform. There is a mixture of past and past perfect tense under the description of the study population and setting.

There is also an extensive use of abbreviations in the manuscript which makes it difficult to comprehend. My suggestion is that the abbreviations should be reduced.

What was the actual sample size used in this study, 2,701 or 1,401??

Tables need to be re-designed to look simple and easy to visualize.

Reviewer #2: Thank you for the article.

16-The numbers and percentages of ever use of PrEP is not matching in line 16 and 17.

122- Since the community is already sensitised about the programme there is awareness on PrEP in the community. However the "knowledge" regarding PrEP will be a broader term to assess using one single question.

Knowledge and awareness are interchangeably used in the discussion- I think same term throughout would be better.

Table 2 and Table 3 are overcrowded with numbers.The number of categories in some of the variables like occupation could be reduced by clubbing.

Regression analysis could be limited to only those variables that are significantly associated with the outcome variable.

All the best!

6. PLOS authors have the option to publish the peer review history of their article (what does this mean? ). If published, this will include your full peer review and any attached files.

**Do you want your identity to be public for this peer review?** For information about this choice, including consent withdrawal, please see our Privacy Policy .

Reviewer #1: No

Reviewer #2: No

---

## [Decision Letter · Decision Letter 1]

17 Jan 2025

Pre-exposure prophylaxis (PrEP) awareness, use, and discontinuation among Lake Victoria fisherfolk in Uganda: a cross-sectional population-based study

PGPH-D-24-01886R1

Dear Kate,

We are pleased to inform you that your manuscript 'Pre-exposure prophylaxis (PrEP) awareness, use, and discontinuation among Lake Victoria fisherfolk in Uganda: a cross-sectional population-based study' has been provisionally accepted for publication in PLOS Global Public Health.

Best regards,

Collins Otieno Asweto, PhD

Academic Editor

Reviewer's Responses to Questions

**Comments to the Author**

1. If the authors have adequately addressed your comments raised in a previous round of review and you feel that this manuscript is now acceptable for publication, you may indicate that here to bypass the “Comments to the Author” section, enter your conflict of interest statement in the “Confidential to Editor” section, and submit your "Accept" recommendation.

Reviewer #3: All comments have been addressed

Reviewer #4: All comments have been addressed

2. Does this manuscript meet PLOS Global Public Health’s publication criteria ? Is the manuscript technically sound, and do the data support the conclusions? The manuscript must describe methodologically and ethically rigorous research with conclusions that are appropriately drawn based on the data presented.

Reviewer #3: Yes

Reviewer #4: Yes

3. Has the statistical analysis been performed appropriately and rigorously?

Reviewer #3: Yes

Reviewer #4: Yes

4. Have the authors made all data underlying the findings in their manuscript fully available (please refer to the Data Availability Statement at the start of the manuscript PDF file)?

Reviewer #3: Yes

Reviewer #4: Yes

5. Is the manuscript presented in an intelligible fashion and written in standard English?

Reviewer #3: Yes

Reviewer #4: Yes

6. Review Comments to the Author

Reviewer #3: Thank you so much, your manuscript reads well after addressing the comments raised earlier on and I recommend it for publication. You may also need to just format the references for the journal requirement, but this ofcourse could be done in the production /proof-reading process. All the best

Reviewer #4: Ths manuscript is an important HIV prevention study. The Authors did an excellent job in their presentations. Adding conversion rate would give more insight to the efficacy of the PrEP program.

7. PLOS authors have the option to publish the peer review history of their article (what does this mean? ). If published, this will include your full peer review and any attached files.

**Do you want your identity to be public for this peer review?** For information about this choice, including consent withdrawal, please see our Privacy Policy .

Reviewer #3: **Yes: ** Moses Banda Aron

Reviewer #4: **Yes: ** Prince Obinna Anyanwu
